# Spare Parts from Discarded Materials: Fetal Annexes in Regenerative Medicine

**DOI:** 10.3390/ijms20071573

**Published:** 2019-03-29

**Authors:** Giulia Gaggi, Pascal Izzicupo, Andrea Di Credico, Silvia Sancilio, Angela Di Baldassarre, Barbara Ghinassi

**Affiliations:** Department of Medicine and Aging Sciences, University “G. D’Annunzio” of Chieti-Pescara, 66100 Chieti, Italy; giulia.gaggi@unich.it (G.G.); izzicupo@unich.it (P.I.); andrea.dicredico@hotmail.it (A.D.C.); silvia.sancilio@unich.it (S.S.)

**Keywords:** perinatal stem cells, amniotic fluid stem cells, amniotic epithelial cells, amniotic mesenchymal cells, chorionic mesenchymal stromal cells, Wharton’s jelly stem cells, decellularized tissues, fetal membranes, regenerative medicine, differentiation

## Abstract

One of the main aims in regenerative medicine is to find stem cells that are easy to obtain and are safe and efficient in either an autologous or allogenic host when transplanted. This review provides an overview of the potential use of the fetal annexes in regenerative medicine: we described the formation of the annexes, their immunological features, the new advances in the phenotypical characterization of fetal annexes-derived stem cells, the progressions obtained in the analysis of both their differentiative potential and their secretoma, and finally, the potential use of decellularized fetal membranes. Normally discarded as medical waste, the umbilical cord and perinatal tissue not only represent a rich source of stem cells but can also be used as a scaffold for regenerative medicine, providing a suitable environment for the growth and differentiation of stem cells.

## 1. Introduction

Stem cell biology has become one of the most widely studied fields of biology, especially in the context of regenerative medicine for the repair and replacement of damaged tissues and organs [1,2].

Based on their origin, stem cells can be classified into four categories. Embryonic, induced pluripotent, perinatal, and adult stem cells. Embryonic stem cells (ESCs), derived from a blastocyst 5–6 days after fertilization, are pluripotent, being able to differentiate into the three germ layers and being capable of self-renewal, but their extended culture time in vitro results in chromosomal abnormality and instability [3]. Induced pluripotent stem cells (iPSCs) were obtained for the first time by Yamanaka et al., who reprogrammed murine and human fibroblasts using the ectopic expression of four transcription factors (OCT-4, SOX-2, Klf4, and c-myc), restoring adult cells to their pluripotent state [4]. Perinatal stem cells can be isolated from the amniotic fluid, placenta, and umbilical cord. These cells cannot divide indefinitely in vitro; however, they are generally considered multipotent, because they can differentiate into a related family of cells. Nevertheless, their real position in the stemness hierarchy is still unclear [5]. Adult stem cells reside within organs during post-natal life. They usually are oligo- or unipotent; thus, they can differentiate into closely related cell types and play a role in tissue regeneration [3]. To date, the highest efficiency of differentiation in tissue-specific cells has been obtained from ESCs and iPSCs, but their use has safety limitations due to their tumorigenic potential. In addition, the isolation of ESCs involves the destruction of a blastocyst, raising ethical concerns [6,7,8]. For this reason, scientists have focused their attention on stem cells that can be isolated from perinatal tissue. Fetal annexes, such as the placenta, umbilical cord, and amniotic fluid, represent an interesting source of stem cells for clinical and research purposes, because they are not tumorigenic and their use does not cause ethical concerns [9]. Moreover, perinatal sources of stem cells have several advantages over adult sources of stem cells (i.e., bone marrow and adipose tissue) in terms of high capacity of proliferation, easy recovery, and availability [10]; indeed, the placenta and umbilical cord are typically discarded after birth, while amniotic fluid cells can be isolated from amniotic fluid routinely obtained during amniocentesis [11]. Although, to date, it is still unclear how the immune system of potential recipients might perceive tissues differentiated from fetal annex-derived stem cells, some studies have predicted that these tissues may not be immunogenic. It has been evidenced, indeed, that fetal stem cells actively exert a local, nonspecific suppressive effect on T lymphocytes finalized to the acceptance of the fetal allograft. 

Materials derived from the extracellular matrix of intact mammalian tissues have been successfully used in a variety of tissue engineering/regenerative medicine applications in both preclinical and clinical applications. Recently, the exploitation of decellularized fetal membranes as biological scaffolds has provided several breakthroughs. 

This review provides an overview of the potential use of fetal annexes in regenerative medicine: we describe the steps related to the formation of annexes, the immunological features that characterize the cells present in fetal membranes, the new advances in the phenotypical characterization of fetal annex-derived stem cells, the progress obtained in the analysis of both their differentiative potential and their secretoma, and finally the exploitation of decellularized fetal membranes in regenerative medicine.

## 2. Development of Embryonic and Fetal Annexes

The blastocyst reaches the lumen of the uterus 4–5 days after fertilization. At the embryonic pole, the cytotrophoblast that surrounded the blastocyst produces a syncytial layer, the syncytiotrophoblast, which invades the uterine epithelium, forming an association with the maternal bloodstream [12]. The development of the uteroplacental circulation is accompanied by the formation of the embryonic and fetal membranes, which include the amnion, chorion, and yolk sac (Figure 1A). Soon after the implantation, the inner cell mass of the blastocyst differentiates into the hypoblast and epiblast. The yolk sac derives from hypoblast cells that migrate along the inner surface of the cytotrophoblast; its primary role is to provide nourishment for the embryo at the earliest stages of development. Then, it becomes reduced in size. At the fourth week of development, during organogenesis, part of the yolk sac is incorporated into the embryo as the gut. 

The amniotic epithelium is derived from the epiblast. Indeed, epiblast cells migrate outwards and generate the amnioblasts; the amniotic cavity forms within 7–8 days after fertilization, and the epiblast cells and amnioblasts form the epithelial cell lining of the amniotic cavity [7]. The amniotic fluid contained in that cavity allows the fetus to freely grow and move inside the uterus [13]. Gastrulation starts between day 9 and day 15 in humans, when the epiblast gives rise to the three primary germ layers of the embryo. Because the amniotic epithelium develops before gastrulation, it has been suggested that pluripotent stem cells derived from the epiblast may be retained in the amnios, even at term pregnancy.

Chorion, the fetal component of the placenta, derives from trophoblast and from extraembryonic mesoderm and contains fetal blood cells [7]. The chorion undergoes rapid proliferation and forms numerous processes, the chorionic villi, which invade and destroy the uterine decidua. As summarized in Figure 1B, the first association with the maternal vasculature is on day 9, when the invading syncytiotrophoblast forms lacunae, blood-filled sinusoidal spaces in the endometrium; this is the first primitive uteroplacental circulation. Subsequently, the proliferating cytotrophoblast projects digitiform extroflexions into the syncytiotrophoblast, forming primary villi connected with maternal blood. When primary villi show a mesenchyme in their middle, they are called secondary villi. By the third week, the fetal blood and circulatory system are forming, and by day 21, extra-embryonic blood vessels go into the villi, making tertiary villi (Figure 1B) [12]. Chorionic villi cover the entire surface of the early conceptus, but as the gestational sac enlarges, villi in the lower part of the sac next to the decidua basalis are compressed and begin to degenerate; the chorion laeve formation is the result of this partial regression and can be observed by 6–8 weeks of gestation. Because the amniotic cavity enlarges, the amnion fuses with the chorion by 17–20 weeks of gestation, producing the amniochorionic membrane that can be observed up to delivery [7,14].

The fully developed placenta has a discoid shape with a diameter of 15–20 cm and a thickness of 2–3 cm. It is a combination of a fetal portion, formed by the chorion, and a maternal portion called the decidua [15].

### Immunological Properties of Fetal Cells

To date it is still unclear how the immune system of potential recipients might perceive tissues differentiated from stem cells: some studies have predicted that the immune response of the recipient may oppose the engraftment and/or the persistence of the stem cells as a consequence of the histocompatibility, whereas other studies have highlighted the capacity of stem cells to actively exert a local, non-specific suppressive effect on T cells. Therefore, the extent of the immunogenicity and the immunomodulatory effects of the stem cell-derived tissues need to be clarified [6]. Some of the benefits obtained by stem cell therapy are not due to the engrafted cells but are most likely attributable to the release of soluble factors, such as growth factors or immunomodulatory molecules (chemokine receptors, chemo-attractant, or cell-adhesion molecules). These bioactive factors are produced by the stem cells themselves or by damaged cells after interaction with the transplanted stem cells, inducing a protective effect by paracrine signaling [16]. This capacity of stem cells to exert an immunomodulatory action has been described in many in vitro and in vivo studies, but unfortunately, the immunological consequences of stem cells transplantation are still uncertain and affect the planning of clinical trials. Fetal cells are not rejected by the maternal immune system; although the mechanisms that underlie the acceptance of the fetal allograft are not completely understood [15], it is well known that the placenta plays an important role in modulating the maternal immune system during pregnancy [17]. The paradox of maternal tolerance was first analyzed by Peter Medawar in 1953, who proposed that (i) there is a physical separation between the fetus and the mother; (ii) the fetus is antigenically immature; and (iii) the mother has immunological inertia [15]. Since then, many mechanisms that contribute to maternal tolerance have been proposed. However, it is still unclear how fetal cells affect lymphocyte activation and proliferation. Some studies have highlighted an important paracrine effect that involves the stem cell secretion of prostaglandin E2, indoleamine 2,3-dioxygenase-mediated tryptophan depletion, transforming growth factor-β1, hepatocyte growth factor, and leukemia inhibitory factor, but the data remain controversial [12,15,18]. It has also been evidenced that soluble HLA-G molecules produced by the placenta induce the apoptosis of activated CD8+ T cells and inhibit CD4+ T cell proliferation, whereas trophoblast cells expressing HLA-G may inhibit NK cells. Moreover, it has been shown that trophoblast cells that express Fas ligand (CD95L) induce the apoptosis of maternal-activated lymphocytes expressing Fas (CD95) [18]. Nevertheless, the most important immune evasion strategy is the absence of classical MHC class II alloantigens on the trophoblast both in humans and rodents [18]. The absence of MHC class II gives fetal cells the potential to escape recognition by alloreactive CD4+ T cells. Current research on the immunomodulatory properties and immunological characteristics of fetal cells encourages the study of their use in regenerative medicine [19].

## 3. Stem Cells in Fetal Annexes

Ethical barriers and clinical concerns have hampered the use of embryonic and fetal stem cells and iPSCs in the medical context. On the other hand, stem cells present in the fetal annexes are easily obtainable in an inexpensive and non-invasive manner from tissues that are discarded at birth. At least in theory, personalized biobanks for newborns may be created in which stem cells recovered postnatally from fetal annexes might be cryopreserved. Among the perinatal stem cells, stem cells from cord blood are the best characterized; umbilical blood, indeed, represents a useful source of hematopoietic stem cells that are clinically used to treat some hematological diseases [20]. As all the stem cells derived from fetal annexes, umbilical cord blood stem cells display a low immunogenicity and do not require perfect human leukocyte antigen (HLA) tissue matching [21]. On the other hand, the stem populations isolated from the placenta and amniotic fluid and from the tissue surrounding the umbilical cord vessels—i.e., Wharton’s jelly—are less characterized. In the following paragraphs, the phenotypical characteristics and the differentiative potential of stem cells present in the amnion, chorion and in mucoid connective tissue of the umbilical cord have been summarized.

### 3.1. Stem Cells from Amnion/Chorion Membrane

Two type of cells can be isolated from the human amniotic membrane: amniotic epithelial cells (hAECs) and amniotic mesenchymal cells (hAMSC). hAECs cells derive from the amnion epithelium: a thin and avascular membrane composed of a single layer of flat, cuboidal, and columnar epithelial cells in contact with amniotic fluid [15]. For hAEC isolation, the amniotic membrane is mechanically separated from the chorion and then digested with trypsin, which induces the release of epithelial cells [15]. These cells express different surface markers, such as CD24, CD9, E-cadherin, integrins α6 and β1, c-met, SSEA3 and 4, TRA1-60, TRA1-81, OCT-4, SOX-2, and NANOG (Table 1). They also express low levels of HLA-A, -B, and -C. hAECs are negative for CD34, CD133, CD117 (c-kit), CCR4, and SSEA-1. Tamagawa et al. concluded that hAECs are pluripotent; indeed, they created xenogenic chimeric embryoid bodies of labelled hAECs and mouse ESCs, and these embryoid bodies gave rise to the formation of primordial organs, such as neural tubes, the lungs, skin, blood vessels, blood cells, the liver, and components of digestive tracts. Fluorescence imaging reveals the presence of hAECs in each of the organoids, evidencing that these stem cells may give rise to cells of all the germ layers in vitro [22]. The multilineage differentiation of hAECs has been demonstrated by several studies (Table 2). Stimulation with activin A and nicotinamide induces the differentiation of hAECs into insulin-secreting pancreatic cells [23,24], but they can be also driven toward hepatic phenotypes, as evidenced by the expression of hepatic markers, such as albumin, α-fetoprotein, and α1-antitrypsin, and by the acquisition of a similar hepatic morphology [25,26]. Jiawen et al. showed that hAECs, when cultured under osteogenic conditions, increase the expression of osteogenic-related genes, such as Runx2, osteopontin, osterix, and collagen type I [27]. hAECs also show characteristics similar to those of neural progenitor cells, as evidenced by the expression of some markers typical of neurons and glial cells, and when transplanted into the brain of a mouse model for Parkinson’s disease, hAECs interfered with the disease progression by releasing catecholamine and neutrophic factors (Table 3) [26]. 

On the other side, hAMSCs can be obtained from the inner mesodermal tissue (the thicker basement membrane and collagen layer) of amniotic membrane; they display fibroblast-like features and express OCT-3/4, Tra1-60, SSEA3/4, KLf4, c-myc, Tra1-81 [28], desimin, vimentin, CD44 CD90, CD73, and CD105 [29] (Table 1). Their differentiation potential toward mesodermal lineages has been described [30] (Table 2). Toda et al. demonstrated the expression of chondrocyte-related genes, including SOX-5, -6, and -9 and MBP-2 and -4, as well BMP receptors. Moreover, the stimulation of hAMSCs with BMP-2 induces the synthesis of collagen type II and aggrecan in vitro. These results have been confirmed by in vivo experiments; indeed, when transplanted with a collagen scaffold into damaged rat bone, hAMSCs underwent morphological changes and started depositing collagen type II fibers [26]. hAMSCs also display a cardiomyogenic potential: freshly isolated cells express the cardiac specific transcription factor GATA4 and cardiac-specific genes, such as myosin light chain, troponin T and troponin I. After stimulation with bFGF or activin A, hAMSCs express the specific cardiac transcription factor Nkx2.5 and the atrial natriuretic peptide. In addition, α-myosin heavy chain (α-MHC) is expressed in hAMSCs after stimulation with activin A but not with bFGF [26,31]. Chorionic mesenchymal stromal cells (hCMSCs) can be isolated from human chorion by collagenase digestion. These cells, as with hAMSCs, display fibroblast-like features and express CD166, CD105, CD90, CD73, CD44, CD29, and CD13 (Table 1). mRNA expression of OCT-4, SSEA3/4, GATA-2, and REX1 has been also detected [7]. hCMSCs can be successfully driven toward mesodermal linages such as osteogenic [32], chondrogenic, and adipogenic differentiation [33] (Table 2). Adipogenic differentiation was induced by a permissive medium composed of isobutylmethylxanthine, indomethacin, dexamethasone, and insulin for 3 days followed by 21 days in an adipogenic maintenance medium composed of DMEM, 10% FBS, and insulin. After the differentiation process, the cells showed intracytoplasmic lipid droplets and increased adipose marker expression (aP2, C/EBPα 1, PPARγ2) [33]. The chondrogenic differentiation was instead obtained by stimulating hCMSCs with BMP-6, dexamethasone, ascorbic acid, L-proline, and insulin–transferrin–selenic acid premix for 21 days. After this treatment, cells became positive for type II collagen, SOX-9, and Aggrecan staining [33]. Osteogenic differentiation was induced by culturing hCMSCs in the presence of β-glycerophosphate, l-ascorbic-acid 2-phosphate, and dexamethasone. At the end of the differentiation process, calcium mineralization was detected, while RT-PCR showed an increase of osteogenic markers, such as ALP, osteocalcin, and COL1A [33]. 

Briefly, two distinct types of stem cells can be isolated from amniotic membrane: hAECs, which derive from the epithelial part of amnion, and hAMSCs, which can be isolated from the mesenchymal layer of amnios. On the other hand, hCMSCs can be isolated from the chorionic membrane. Thus, manually separating the amnios from the chorion after the delivery, it is possible to isolate hAECs, hAMSCs, or hCMSCs: these three populations differ with respect to morphology, phenotype, and differentiative potential (Table 1 and Table 2).

### 3.2. Wharton Jelly-Derived Stem Cells

The umbilical cord is an extra-embryonic tissue, linking mother and fetus. In the umbilical cord, there are two arteries and one vein surrounded by a mucoid connective tissue, called Wharton’s jelly, composed of myofibroblast-like stromal cells, proteoglycans, and collagen fibers [34]. Cells that can be isolated from Wharton’s jelly have mesenchymal features and can be cultured for more than 80 population-doublings with no changes in morphology or indication of senescence [34]. They express CD73, CD90, CD105, the stemness marker OCT-3/4, Nanog, and SOX-2 [34,35] (Table 1). As stem cells from amniotic membrane, Wharton’s jelly stem cells (WJ-SCs) show immunomodulatory activity. Troyer et al. showed that there was no immune rejection of undifferentiated WJ-SCs in vivo and that they were well tolerated in an allogenic transplantation [58]. WJ-SCs may have a differentiation potential into several lineages (Table 2). Bask et al. demonstrated that WJ-SCs have osteogenic potential and may be a cell source for bone repair in regenerative medicine [43]. More recently, Ahmadi and colleagues used WJ-SCs to coat a poly-l-lactic acid electrospun nanofiber (PLLA) scaffold for bone tissue engineering applications [59]. It has been also demonstrated that, in undifferentiated conditions, WJ-SCs express neural markers, such as nestin. After neurogenic stimulation, these cells undergo changes in morphology and express high levels of neuronal proteins, such as neuron-specific enolase (NSE), 2’3’-cyclic nucleotide-3’-phospodiesterase (CNPase), and glial fibrillary acidic protein (GFAP) [34]. Recently, Raut et al. showed that WJ-SCs can differentiate into a hepatic lineage through induction with Valproic acid for 48 h, followed by the sequential addition of hepatic differentiation medium, composed of 2% FBS, FGF-4, and HGF, for 14 days. Cell maturation is then achieved with a cocktail of dexamethasone, HGF, transferrin, insulin, and selenous acid for 13 days. At the end of the culture, cells display features similar to mature hepatocytes, such as polygonal shape morphology, the presence of multinucleated cells, and the expression of hepatic markers, such as albumin, cytokeratin18, α-fetoprotein, hepatocyte nuclear factor 4α, anti-trypsin, and cytochrome P450 3A4 [48]. These differentiated hepatocyte-like cells also develop the ability to store glycogen, to produce urea, and to release albumin. Furthermore, Raut et al. identified a signature of 63 significantly expressed miRNAs during hepatic differentiation; the putative target genes of these miRNAs appeared to be mostly associated with essential processes for hepatic differentiation, cholesterol, and drug metabolic processes and stemness maintenance [48]. It has been shown that WJ-SCs are able to differentiate into retinal progenitor cells in vitro using a medium supplemented with Dkk-1 and LeftyA, which act as Wnt/β catenin and nodal signaling pathway antagonists, respectively. After the differentiation process, cells express the retinal progenitor markers Pax6 and Rx [50].

### 3.3. Amniotic Fluid-Derived Cells

Amniotic fluid, which is easily accessible with amniocentesis, contains cells that are derived from the fetus. The first evidence that amniotic fluid could contain pluripotent stem cells was provided by Prusa et al. when they described the presence of a sub-population of proliferating amniotic fluid cells expressing the pluripotency marker OCT-4 both at the transcriptional and protein level. Antonucci et al. showed that amniotic fluid-derived stem cells (AFSCs) express genes known to be markers of pluripotency, such as Nanog, OCT-4, C-Myc, KLF4, c-kit and Ovol1, Stella, and Fragilis [5]. The surface antigenic profile of AFSCs includes embryonic and mesenchymal markers, such as SSEA-4, CD73, OCT-4, CD90, and CD105 (Table 1). AFSCs maintain a constant morphology and marker and gene expression up to 25 passages. Moreover, they do not form tumors when transplanted into SCID mice [13]. Their pluripotency has been established on the basis of their ability to form embryoid bodies and on the observation that they can differentiate into cells from three germ layers [13] (Table 2). Spitzhorn et al. observed a successful differentiation of AFSCs into adipogenic, osteogenic, and chondrogenic lineages. Indeed, results showed that after the adipogenic differentiation process, cells were positive for Oli Red O solution, which marks fat vacuoles. During chondrogenic differentiation, alcian blue staining showed the presence of proteoglycans, whereas the osteogenic differentiation was assessed by alizarin red S staining, which marks calcium deposits [11]. Maraldi et al. guided the c-kit^+^ AFSCs towards glial differentiation using two different methods, one based on retinoic acid and DMSO administration and one in which the cells were exposed to β-mercaptoethanol, DMSO, and butylated hydroxyanisole. Both protocols led to an increase of glial markers, such as the glial fibrillary acidic protein (astrocytic marker), the peripheral myelin protein 22, the myelin-associated enzyme CNPase (oligodendrocytic marker), and S-100, a protein that is normally expressed by cells from the neural crest, such as melanocytes, glial cells, and Schwann cells [54]. The authors tried also to obtain neuron-like cells. After three weeks of culture, c-kit^+^ AFSCs acquired a typical neuronal morphology and about 30% of cells expressed neuronal markers, such as β-III tubulin, NeuN, MAP2 (a neuron-specific cytoskeletal protein involved in microtubule assembly), and synapsin. In any case, the authors showed that only a few AFSC samples possessed a proliferation rate sufficiently high to tolerate the neuronal differentiation conditions [54]. AFSCs’ differentiation toward oligodendrocytes and neurons was also observed in in vivo experiments [54]. Carnevale et al. obtained insulin-producing β-cells from c-kit^+^ AFSCs; after 14 days of culture in the presence of trans-retinoic acid, β-mercaptoethanol, and nicotinamide, cells increased the expression of PDX-1, insulin, and Glut2 [56]. Recently, Di Baldassarre et al. demonstrated that unselected AFSCs might differentiate toward the cardiac lineage, giving rise to CM-like cells characterized by several cardiac-specific molecular, structural, and functional properties. In particular, these authors showed that the subset of human amniotic fluid samples expressing the multipotency markers SSEA4, OCT-4, and CD90 (^Cardiopoietic^AF cells) responded to the differentiation process by increasing the expression of the cardiac transcription factors Nkx2.5 and GATA4, sarcomeric proteins (cTnT, α-MHC, α-SA), Connexin43, and atrial and ventricular markers. Furthermore, differentiated cells were positive for pumps and involved the excitation/contraction coupling of CACNA1C, SERCA, Nav1.5, KCNQ1, and Kir2, with approximately 30% of ^Cardiopoietic^AF-derived CM-like cells responding to caffeine or adrenergic stimulation [53]. AFSCs also have a differentiation potential in hepatic lineage. Stimulation with HGF, oncostatin, and dexamethasone increased the expression of cytokeratin 18 and α-fetoprotein [55].

## 4. Paracrine Effect of Fetal Annex-Derived Stem Cells

Evidence has shown that stem cells release trophic signals that can alter the local microenvironment in injured tissue, inducing the repair of damage. Indeed, the beneficial effects obtained following an injection of stem cells into an injured organ seem to be also (or mainly) due to their modulatory paracrine effects. Growth factors, cytokines, RNA, and all molecules released from stem cells are called “secretomes”. In this scenario, scientists have focused on the characterization of stem cell-secreted extracellular vesicles (EVs), membrane-bound cellular components enriched with soluble and bioactive factors that play an important role in cell-to-cell communication [60]. According to their size, EVs can be divided into microvesicles (MVs) and exosomes [61]. MVs have a diameter of 100–1000 nm and are formed by the external budding of cell membranes [61]. MVs encompass high quantities of phosphatidylserine-containing proteins associated with lipid rafts and are enriched with surface markers, such as CD40, sphingomyelin, cholesterol, and ceramide. MVs interact with target cells by specific receptor–ligand interactions [61]. Exosomes have a diameter of 40–100 nm and contain large amounts of annexin, CD63, CD9, CD81, and heat-shock proteins, such as Hsp 60, Hsp70, and Hsp90; they also express clathrin and tumor susceptibility genes. The membrane that encapsulates exosomes is rich in cholesterol, sphingolipids, and ceramide, but, unlike MVs, it contains small amounts of phosphatidylserine [61]. Exosomes interact with recipient cells through three different pathways: (i) they can transmit the contents through adhesion to cell surfaces via lipid–ligand receptor interaction; (ii) exosomes may enter into cells via endocytic uptake; and (iii) exosomes may enter into cells by the fusion of the vesicles with the target cell membrane [61]. Evidence about the important role of secretome in cellular communication prompted studies on the biomolecules released by fetal annex-derived stem cells. By genomic and proteomic approaches, the ability of the perinatal stem cells to produce a wide spectrum of bioactive molecules, such as cytokines, chemokines, growth factors, extracellular matrix components, and miRNA (summarized in Table 3) has been demonstrated.

### 4.1. Paracrine Effect of hAECs

Song et al. analyzed cytokine release from hAECs and demonstrated the secretion of angiogenin, interleukin (IL-6), monocyte chemoattractant protein, and epidermal growth factor (EGF). These cytokines release in significantly increased quantities when cells are cultured in hypoxic conditions. Moreover, when transplanted in animal models of cardiac ischemic damage, hAECs release bioactive molecules, as demonstrated by the presence of human angiogenin, IL-6, monocyte chemoattractant protein, and EGF in the infarcted area of rat myocardium; this evidence suggests that the paracrine effect of hAECs might participate in cardiac regeneration [62]. Recently, Zhao et al. demonstrated in vitro that exosomes from hAECs promote the proliferation and migration of human fibroblasts and that this effect is mediated by the RNA components of hAEC exosomes; indeed, this response was abolished by treatment with RNase. Moreover, in vivo experiments evidenced that hAEC-derived exosomes ameliorate the arrangement of collagen fibers during wound repair in male BALB/c mice [68]. hAEC-conditioned medium (hAEC-CM) affects the metabolism of FOB1.19, a cell line of human fetal osteoblasts used to study the osteogenic differentiation process, by increasing both its proliferation and migration rate [63]. Moreover, when FOB1.19 underwent an osteogenic differentiation process in the presence of hAEC-CM, a higher expression of osteogenic differentiation-related markers, such as alkaline phosphatase, osteocalcin, and osteopontin, was detected. Further analysis revealed that the supernatant of hFOB1.19 treated with hAEC-CM contained higher concentrations of TGFβ1 and miR-34a-5p, suggesting that these two factors may contribute to osteogenic differentiation [63].

### 4.2. Paracrine Effects of Wharton’s Jelly Stem Cells

WJ-SCs are able to secrete high levels of chemokines and growth factors, which create a chemoattractive environment, and WJ-SC-CM promotes immunosuppressive, antioxidant, and proliferative effects [64]. Zhou et al. demonstrated that MVs isolated from hWJ-SCs and intravenously administrated in rats after a kidney ischemia reduce tubular lesions, the apoptotic rate, and fibrosis [69]. Immunohistochemistry also evidenced that MV treatment reduces the infiltration of macrophages inside injured kidney tissues; an effect on IL-10 and on TNF-α secretion was also observed [69]. These results suggest that WJ-SC-derived MVs attenuate the inflammatory response to ischemic renal damage [69]. Other studies showed that hWJ-SC transduced with the *Wnt7* gene released Wnt in the supernatant, increasing the migration of fibroblasts in the damaged area and enhancing the expression of extracellular matrix components (collagen I and III and α-SMA) in the wound area of mice [70]. Moreover, Dong et al. showed that, during re-epidermalization, mice treated with Wnt-enriched CMs have a thicker epidermis with more organized cell layers in comparison with the control group. On the contrary, keratinocyte migration was significantly retarded by Wnt-enriched CMs 24 h after scratching [70].

### 4.3. Paracrine Effects of Amniotic Fluid Cells

Transmission electron microscopy analysis revealed that hAFSC can secrete EVs ranging in size from 50 to 1000 nm [60]. Western blot analysis confirmed that AFSC-derived MVs and EVs express the canonical markers TSG and ALIX, whereas cytometry evidenced a significant subpopulation of exosomes expressing CD81, CD9, AnnV, and CD63 [60]. In particular, when hAFSCs were exposed to a 24 h preconditioning (serum-free and hypoxic conditions), a significant enrichment of CD81+, CD9+, and CD63+ exosomes compared with the normoxic condition was observed [60]. Balbi et al. analyzed the role of hAFSC-derived EVs on different target cells. Data from C2C12 cells evidenced that hAFSC-derived EVs mediate a dose-dependent antiapoptotic effect and significantly reduce the oxidative damage by H2O2 by improving both the cell viability and proliferation [60]. The proangiogenic effects of hAFSC-EV were then analyzed in a mouse Matrigel plug assay; in this in vivo model, hAFSC-EV determined an upregulation of vascular markers VEGFA and Pecam1 by the endothelial cell colonizing the plug [60]. An immunomodulatory effect for hAFSC-derived EVs has also been described; although human peripheral mononuclear cells internalized EVs with a lower efficiency than other cell types, hAFSC-derived EVs reduced the maturation of CD27+CD19+ memory B cells in response to pokeweed mitogens [60] and the percentage of T-cells in S and G2/M phases [45]. In a mouse model of skeletal muscle atrophy, hAFSC-derived EVs reduced IgG and CD68+ cell infiltration with a significant decrease of IgG-damaged fibers. This effect was accompanied by a modification of the biomolecular milieu as evidenced by both the decrease of proinflammatory IL-1α and IL-4 and the upregulation of anti-inflammatory cytokine IL-10 [60]. hAFSC-derived EVs contain small non-coding RNAs and miRNAs that can be released into target cells. In particular, real-time PCR analysis revealed the significant enrichment of some regenerative miRNAs carried by hAFSCs EVs [60].

## 5. Fetal Annexes as a Scaffold in Regenerative Medicine

Scaffolds create an appropriate support by generating a three-dimensional environment for stem cell attachment, growth, and differentiation [71]. They can efficiently integrate into the host when the tissue is transplanted in vivo [72]. Recently, in the field of regenerative medicine, scientific attention has been focused on scaffolds derived from decellularized tissues as an alternative to synthetic polymers [73,74]. The use of fetal membrane in skin transplantation was first reported by Davis in 1910. Since then, human-derived placental tissues have been used to improve healing by means of their immunomodulating and antibacterial properties and by their ability to support the migration, proliferation, and functional maturation of cells [74]. Decellularized amnion and chorion membranes have been utilized in the medical approach to diabetic or venous ulcers [75]. Some authors have shown that these placental derived membranes are also able to evoke paracrine signals that support the healing and tissue repair in allograft recipients [76]. Dorazehi et al. evidenced that the decellularized human amniotic membrane may be a suitable scaffold for neuronal differentiation. Indeed, they seeded murine bone-marrow mesenchymal stem cells (BM-MSCs) on a decellularized human amniotic membrane and cultured them with embryonic rat cerebrospinal fluid. After 7 days, the cells expressed two neuronal markers, MAP-2 and β-tubulin III [74]. Amniotic membranes can also be used as a support system for human chondrocytes in human articular cartilage repair. Human chondrocytes can be cultured on the chorionic basement side of the amniotic membrane (they do not grow on the epithelial side); in vitro experiments evidenced that, in this condition, human chondrocytes create a new tissue expressing type II collagen that integrates with osteoarthritis cartilage, providing a superficial cell cover that decreased the irregularities of the damaged cartilage. Moreover, chondrocytes from the amniotic membrane migrate and penetrate into the cavities of the cartilage, cancelling the boundary between native and new cartilage [77]. Francisco et al. used amniotic membrane as a scaffold for the repair of pericardium in rats. They substituted a segment of pericardium with a patch of human decellularized amniotic membrane. In the 30-day postoperative period, no graft-related morbidity (infections, graft failure) was observed, and no calcification occurred in the area. It is known that calcification and degeneration processes are limiting factors in the longevity and clinical effectiveness of biomaterial. On the contrary, in these experiments, amniotic membrane patches were fully integrated into the host myocardium, suggesting that amniotic membrane can be a potential biomaterial candidate for this application. Indeed, 4 weeks after surgery, host cells organized with tissue fibrils and capillaries were clearly identified in the epicardial layer coating the amniotic membrane patch [78]. In addition, decellularized Wharton’s jelly has been studied as a scaffold in regenerative medicine. WJ is a mucoid connective tissue surrounding umbilical cord vessels. It contains an extracellular matrix composed of type I, II, IV, and V collagen, hyaluronic acid, and glycosaminoglycans (GAGs); in addition, it is a reservoir of aFGF, bFGF, IGF-I, EGF, PDGF, and TGF-β1. Because type I collagen is very abundant in WJ, it appears to be an ideal candidate for the reconstruction of collagen-based tissue, such as skin, bone, and cartilage [79]. It has been demonstrated that the spongy scaffold obtained from the decellularized WJ extracellular matrix can be used as a support for human fibroblasts that proliferate and penetrate into the three-dimensional (3D structure of the WJ scaffold; for this reason, it can be useful for the reconstruction of damaged skin in a mouse full-thickness wound model [79]. Indeed, Beiki et al. demonstrated that wound exudates disappeared more promptly and the wound area reduced more rapidly in a scaffold-treated mice group. Moreover, complete re-epithelialization, epidermal ridges, and primary hair follicles were observed in the scaffold-treated group. On day 18, full reepithelization, a developed dermis, and newly formed blood vessels in the hypodermis were seen only in the scaffold-implanted group [79]. Moreover, a decellularized WJ matrix might be used as a scaffold for WJ-SCs or bone marrow mesenchymal stem cells; in particular, Jadalannagari et al. implanted this matrix into a murine calvarial defect model, which was able to attract in vivo GFP-labelled osteocytes [78].

## 6. Conclusions

The application of stem cells in regenerative medicine has some important obstacles, because they should be easily obtainable and safely and efficiently transplantable. Stem cells from perinatal tissues are accessible, show low immunogenicity, and do not raise ethical concerns. All these characteristics support their potential use in regenerative medicine and encourage the translation of preclinical studies into therapy. Normally discarded as medical waste, the umbilical cord and perinatal tissue not only represent a rich source of stem cells but can also be used as a scaffold for regenerative medicine, providing a suitable environment for the growth and differentiation of stem cells.

## Figures and Tables

**Figure 1 ijms-20-01573-f001:**
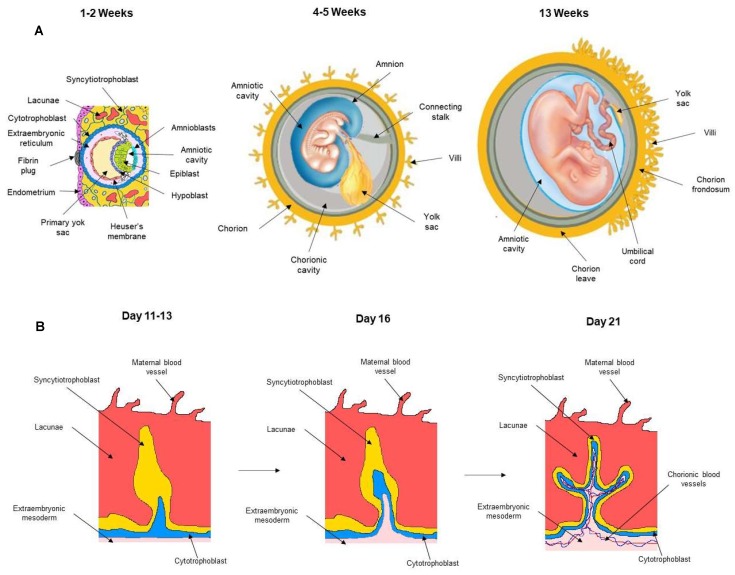
Uteroplacental circulation and fetal annex formation. (**A**) Development of embryonic and fetal annexes. (**B**) Development of chorionic villi. The scheme represents the formation of the primary, secondary, and tertiary villi.

**Table 1 ijms-20-01573-t001:** Phenotype and morphology of stem cells from fetal annexes.

Cell Type	Phenotype			Morphology
		Markers	References	
hAECs	Embryonic cells	Nanog	[1,7]	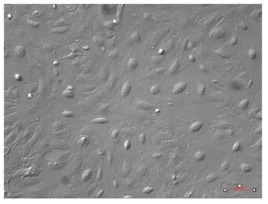
		OCT-4		
		SOX-2		
	Pluripotency	Tra1-60		
		Tra1-81		
		SSEA3		
		SSEA4		
	Mesenchymal	CD24		
	Endothelial cells	E-cadherin		
		Integrin α6		
		Integrin β1		
	Immune cells	CD9		
hAMSCs	Embryonic cells	OCT-3	[28,29]	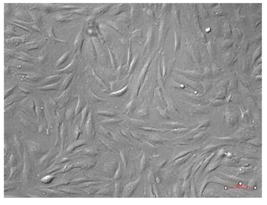
		OCT-4		
		Klf4		
		c-myc		
	Pluripotency	Tra1-60		
		Tra1-81		
		SSEA3		
		SSEA4		
	Mesenchymal	CD90		
		CD24		
		Vimentin		
		CD73		
	Hematopoietic	CD44		
		CD105		
	Muscle tissue	Desmin		
hCMSCs	Embryonic cells	OCT-3	[7]	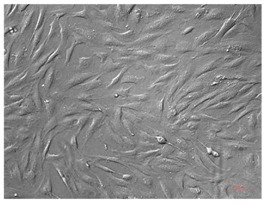
	Pluripotency	SSEA3/4		
		REX1		
	Mesenchymal	CD90		
		CD73		
		CD29		
	Hematopoietic	CD44		
		CD105		
		CD19		
hWJ-SCs	Embryonic cells	Nanog	[34,35]	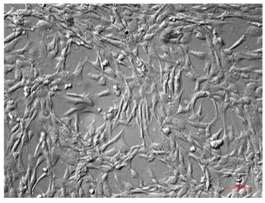
		OCT-3		
		OCT-4		
		SOX-2		
	Mesenchymal	CD90		
		CD73		
	Hematopoietic	CD105		
hAFSCs	Embryonic cells	Nanog	[5,13]	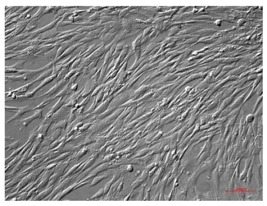
		OCT-4		
		Klf4		
		SOX-2		
	Pluripotency	Stella		
		Fragilis		
		Ovol1		
		SSEA4		
		c-myc		
		c-kit		
	Mesenchymal	CD90		
		CD73		
	Hematopoietic	CD105		

Legend: hAECs, human amniotic epithelial cells; hAMSCs, human amniotic mesenchymal stem cells; hCMSCs, human chorionic mesenchymal stromal cells; hWJ-SC, human Wharton’s jelly stem cells; hAFSCs, human amniotic fluid stem cells. Cells have been observed with AxioVert A1 using the plasDIC plan-neofluar 10X/0,3 M27; objective Images have been acquired with a Camera Axiocam 503 Mono and analyzed with ZEN Software (Carl Zeiss, Jena, Germany). Scale bar 100 μm.

**Table 2 ijms-20-01573-t002:** Differentiative potential of fetal annex-derived stem cells.

Cell Type	Differentiative Potential							
	*Chondrogenic*	*Adipogenic*	*Osteogenic*	*Cardiac*	*Neuronal*	*Hepatic*	*Pancreatic*	*Retinal*
hAECs	[26,36]	[36]	[27]	[37]	[36]	[25,26,38]	[23,24]	
hAMSCs	[39]	[33,40]	[41]	[26,42]				
hCMSCs	[33]	[33]	[32]					
hWJ-SCs	[34,43]	[43]	[43,44]	[45]	[34,46,47]	[48]	[49]	[50]
hAFSCs	[11,51]	[11,51]	[11,51,52]	[53]	[54]	[55]	[56,57]	

Legend: hAECs, human amniotic epithelial cells; hAMSCs, human amniotic mesenchymal stem cells; hCMSCs, human chorionic mesenchymal stromal cells; hWJ-SC, human Wharton’s jelly stem cells; hAFSCs, human amniotic fluid stem cells.

**Table 3 ijms-20-01573-t003:** Secretome of fetal annex-derived stem cells.

Cell Type	Secretome		Ref.
hAECs	Cytokines	ANG	[17,62]
		EFG	
		IL-6	
		MCP-1	
		Dopamin	
		NGF	
		Neurotrophin-3	
		BDNF	
	miRNAs	miR-34a-5p	[63]
hWJ-SCs	Cytokines	EGF	[64,65]
		IP-10	
		ANG	
		BDNF	
		SDF-1	
		IGF	
		TGF-α	
		TGF-β	
		HGF	
		VEGF	
		VEGF-R2	
		FGF-b	
		PDGF-BB	
		IL-8	
		IL-6	
		IL-10	
		VCAM-1	
		MCP-1	
		SCF	
	miRNAs	miR-7-5p	[66]
		miR-10a-5p	
		miR-99a-5p	
		miR-100-5p	
		miR142-3p	
		miR-144-3p	
		miR-196a-5p	
		miR-452-5p	
hAFSCs	Cytokines	SFD-1a	[67]
		PDGF-BB	
		VEGF	
		LIF	
		B-NGF	
		SCF	
		FGF-b	
		HGF	
	miRNA	miR-223	[60]
		miR-146a	
		miR-let 7c	
		miR-21	
		miR-126	
		miR146b	
		miR-199a-3p	
		miR2-210	

Legend: hAECs, human amniotic epithelial cells, hWJ-SC, human Wharton’s jelly stem cells, hAFSCs human amniotic fluid stem cells.

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
