# Peer review of "Spare Parts from Discarded Materials: Fetal Annexes in Regenerative Medicine"

_ijms, 2019, doi:10.3390/ijms20071573_

Reviewer 1 Report

In the present manuscript, Gaggi and co-authors described the remarkable characteristics of placenta-derived cells, in particular, the potential use of the fetal annexes in regenerative medicine. The authors here delineated the properties of perinatal tissues, representing them both as a rich source of stem cells and as a scaffold for supporting stem cells characteristics.

In order to achieve the merit of publication, I would suggest authors provide  a minor revision in order to argue the following points:

Minor comments:

1) Please, in the section “Development of embryonic and fetal annexes”, line 63, correct the gastrulation start period in human from day 9 to day 15;

2) Please, in the section “Development of embryonic and fetal annexes”, line 76, correct the contribution of embryonic blood vessels into extra-embryonic blood vessels;

3) Please, in the section “3.1 Stem cells from amnion/chorion membrane ”, line 185, consider the most common nomenclature reported in the literature for chorionic mesenchymal stromal cells, indicated as hCMSCs instead of hCSCs;

4) Please, in table 2, verify and consider to add the references about the pancreatic differentiation potential of hWJ-SCs;

5) In the legend of table 3, the hAMCSs are not reported in the table but cited in the legend: please, check and correct the sentence; 

Author Response

Reviewer 1

In the present manuscript, Gaggi and co-authors described the remarkable characteristics of placenta-derived cells, in particular, the potential use of the fetal annexes in regenerative medicine. The authors here delineated the properties of perinatal tissues, representing them both as a rich source of stem cells and as a scaffold for supporting stem cells characteristics.

In order to achieve the merit of publication, I would suggest authors provide  a minor revision in order to argue the following points:

Minor comments:

1) Please, in the section “Development of embryonic and fetal annexes”, line 63, correct the gastrulation start period in human from day 9 to day 15; 

We agree with the reviewer and apologize for the imprecision. We changed the sentence, according to reviewer’ suggestion.

2) Please, in the section “Development of embryonic and fetal annexes”, line 76, correct the contribution of embryonic blood vessels into extra-embryonic blood vessels; 

We apologize for this material error. We thank the reviewer for the report. The sentence has been now revised according to the reviewer’s suggestion.

3) Please, in the section “3.1 Stem cells from amnion/chorion membrane ”, line 185, consider the most common nomenclature reported in the literature for chorionic mesenchymal stromal cells, indicated as hCMSCs instead of hCSCs; 

Done.

4) Please, in table 2, verify and consider to add the references about the pancreatic differentiation potential of hWJ-SCs; 

Done. We thank the reviewer for the indication. The reference «Pancreatic endocrine-like cells differentiated from human umbilical cords Wharton’s jelly mesenchymal stem cells using small molecules: SHIVAKUMAR et al.», Journal of Cellular Physiology, vol. 234, n. 4, pagg. 3933–3947, apr. 2019” has been added.

5) In the legend of table 3, the hAMCSs are not reported in the table but cited in the legend: please, check and correct the sentence;  

We thank the reviewer for the suggestion. The hAMCSs in the legend was a misprint and it has been erased.

Reviewer 2 Report

The authors provide an overview about the possibility to use the fetal annexes in regenerative medicine.  

The review is consistent and well subdivided into the topics. Notwithstanding this, the authors must look over the gastrulation time reported in the second paragraph and standardize the acronyms as published in the literature (for example hC-MSCs instead of hCSCs). Moreover, in the table 2 the reference 12  do not correspond to the pancreatic differentiative potential.

Author Response

The authors provide an overview about the possibility to use the fetal annexes in regenerative medicine.  

The review is consistent and well subdivided into the topics. Notwithstanding this, the authors must look over the gastrulation time reported in the second paragraph  and standardize the acronyms as published in the literature (for example hC-MSCs instead of hCSCs). Moreover, in the table 2 the reference 12  do not correspond to the pancreatic differentiative potential.

 We thank the reviewer for the suggestion. We corrected both the gastrulation time (“…between day 9 and day 15..”) and all the acronyms, as suggested. In table 2 the previous reference 12 has been replaced with the correct one (B. Okere et al., «In vitrodifferentiation of human amniotic epithelial cells into insulin-producing 3D spheroids», International Journal of Immunopathology and Pharmacology, vol. 28, n. 3, pagg. 390–402, set. 2015)

Reviewer 3 Report

General comments

The submitted manuscript consists in a review focused on the potential use of the fetal annexes in regenerative medicine, covering all the involved aspects.

The subject of this work matches the aim and scope of IJMS, and the topic is worthy of investigation

Minor comments and remarks are listed below point by point.

Keywords

The chosen keywords (i.e. Stem cells, differentiation, regenerative medicine, umbilical cord, perinatal stem cells, amniotic fluid, fetal annexes) should be reported in a more logical order.

1. Introduction

- The introduction section is well organised, but more recent literature references have to be added and a deeper comparison with the  other kinds of stem cells have to be added. -

-  The following statement “Stem cell biology has become one of the most appealing fields of biology, especially in the context of regenerative medicine for the repair and replacement of damaged tissues and organs [1]” has to be corroborated with more recent literature references, including the paper recently published in International Journal of Molecular Sciences , titled Hydrogen Sulfide-Releasing Fibrous Membranes: Potential Patches for Stimulating Human Stem Cells Proliferation and Viability under Oxidative Stress, International Journal of Molecular Sciences 19(8) (2018): 2368.

- Similarly the following sentences “To date, the highest efficiency of differentiation in tissue-specific cells has been obtained from 28 embryonic stem cells (ESCs) and induced pluripotent stem cells (iPSCs), but their use has safety limitations due to their tumorigenic potential” have to be supported with suitable literature references, including “Tuning multi-/pluri-potent stem cell fate by electrospun poly(L-lactic acid)-calcium-deficient hydroxyapatite nanocomposite mats, Biomacromolecules 13 [5] (2012): 1350-1360,”.

 -          Please, specify T cells the first time you used them.

 5. Fetal annexes as a scaffold in regenerative medicine

- The following statements “Scaffolds create an appropriate support by generating a three-dimensional environment for stem cell attachment, growth and differentiation.” and “They can efficiently integrate into the host when the tissue is transplanted in vivo” have to be supported with suitable literature references, including “Design, production and biocompatibility of nanostructured porous HAp and Si-HAp ceramics as three-dimensional scaffolds for stem cell culture and differentiation, Ceramics Silikaty 54[2] (2010): 90-96.”

Author Response

Reviewer 3

 General comments

 The submitted manuscript consists in a review focused on the potential use of the fetal annexes in regenerative medicine, covering all the involved aspects.

The subject of this work matches the aim and scope of IJMS, and the topic is worthy of investigation

Minor comments and remarks are listed below point by point.

Keywords

The chosen keywords (i.e. Stem cells, differentiation, regenerative medicine, umbilical cord, perinatal stem cells, amniotic fluid, fetal annexes) should be reported in a more logical order.

We agree with the reviewer’ suggestion. Keywords have been revised and presented in a more logical order.

1. Introduction

- The introduction section is well organised, but more recent literature references have to be added and a deeper comparison with the  other kinds of stem cells have to be added. –

-  The following statement “Stem cell biology has become one of the most appealing fields of biology, especially in the context of regenerative medicine for the repair and replacement of damaged tissues and organs [1]” has to be corroborated with more recent literature references, including the paper recently published in International Journal of Molecular Sciences , titled “Hydrogen Sulfide-Releasing Fibrous Membranes: Potential Patches for Stimulating Human Stem Cells Proliferation and Viability under Oxidative Stress, International Journal of Molecular Sciences 19(8) (2018): 2368.” 

- Similarly the following sentences “To date, the highest efficiency of differentiation in tissue-specific cells has been obtained from 28 embryonic stem cells (ESCs) and induced pluripotent stem cells (iPSCs), but their use has safety limitations due to their tumorigenic potential” have to be supported with suitable literature references, including “Tuning multi-/pluri-potent stem cell fate by electrospun poly(L-lactic acid)-calcium-deficient hydroxyapatite nanocomposite mats, Biomacromolecules 13 [5] (2012): 1350-1360,”. 

We thank the reviewer for the suggestions.The Introduction section has been revised and improved and more recent references has been added, according to reviewer comment. 

In particular:

-       The introduction has been modified as follow:

Based on the origin, stem cells can be classified into four categories. Embryonic-, induced pluripotent-, perinatal- and adult stem cells. Embryonic stem cells (ESCs) derived from a blastocyst at 5-6 days after fertilization, are pluripotent being able to differentiate into the three germ layers and can self-renew, but their extended culture time in vitro results in chromosomal abnormality and instability [3]. Induced pluripotent stem cells (iPSCs) were obtained first by Yamanaka et al. that reprogrammed murine and human fibroblast using the ectopic expression of four transcription factor (Oct4, Sox2, Klf4 and c-Myc) restoring the adult cells to their pluripotent state [4]. Perinatal stem cells can be isolated from amniotic fluid, placenta and umbilical cord. These cells cannot divide indefinitely in vitro; they are generally considered multipotent since can differentiate into a related family of cells, but their real position in the stemness hierarchy is still unclear. [5]. Adult stem cells reside within organs during post-natal life. They usually are oligo- or unipotent thus they can differentiate into closely related cell types and play a role in tissue regeneration[3]To date, the highest efficiency of differentiation in tissue-specific cells has been obtained from ESCs and iPSCs, but their use has safety limitations due to their tumorigenic potential. In addition, the isolation of ESCs involves the destruction of a blastocyst, raising ethical problems[6],[7],[8]For this reason, scientists have focused their attention on stem cells that can be isolated from perinatal tissue. Fetal annexes such as the placenta, umbilical cord and amniotic fluid represent an interesting source of stem cells for clinical and research purposes, since they are not tumorigenic and their use does not cause ethical concerns [9].Moreover ,perinatal source of stem cells, have several advantages over adult sources of stem cells (i.e. bone marrow, adipose tissue) in terms of high capacity of proliferation, easy recovery and availability [10]; indeed placenta and umbilical cord are discarded after birth, whereas amniotic fluid cells can be isolated from amniotic fluid routinely obtained during amniocentesis [11].

- The suggested references have been added:

 [2]“Hydrogen Sulfide-Releasing Fibrous Membranes: Potential Patches for Stimulating Human Stem Cells Proliferation and Viability under Oxidative Stress, International Journal of Molecular Sciences 19(8) (2018): 2368.” 

[8] “Tuning multi-/pluri-potent stem cell fate by electrospun poly(L-lactic acid)-calcium-deficient hydroxyapatite nanocomposite mats, Biomacromolecules 13 [5] (2012): 1350-1360”

-          Please, specify T cells the first time you used them. 

“T cells” is not an acronym but a common way to indicate T lymphocytes; in the revised version we substituted it with “T lymphocytes”.  

 5. Fetal annexes as a scaffold in regenerative medicine

- The following statements “Scaffolds create an appropriate support by generating a three-dimensional environment for stem cell attachment, growth and differentiation.” and “They can efficiently integrate into the host when the tissue is transplanted in vivo” have to be supported with suitable literature references, including “Design, production and biocompatibility of nanostructured porous HAp and Si-HAp ceramics as three-dimensional scaffolds for stem cell culture and differentiation, Ceramics Silikaty 54[2] (2010): 90-96.” 

We thank the reviewer for the suggestion. The sentences were enriched with appropriate and recent references, as suggested (Rana D et al. Development of decellularized scaffolds for stem cell-driven tissue engineering. J Tissue Eng Regen Med. 2017 Apr;11(4):942-965. doi: 10.1002/term.2061 and Yesmin et al. Bio-scaffolds in organ-regeneration: Clinical potential and current challenges. Curr Res Transl Med. 2017 Sep;65(3):103-113. doi: 10.1016/j.retram.2017.08.002) 

Reviewer 4 Report

The authors provide a comprehensive overview on the potential of using fetal annexes, commonly discarded as medical waste, as source of stem cells for regenerative medicine. The immunological features of the fetal annexes, the phenotypical characterization, differentiation potential and secretomes of the stem cells derived from these annexes are described.

 No major weakness detected

 Specific comments

Please provide a more clear description of Chorionic stroma cells (hCSCs) vs. hAMSCs and add hCSCs to Table 1 and 2

Add A and B to Fig.1

Author Response

Reviewer 4

The authors provide a comprehensive overview on the potential of using fetal annexes, commonly discarded as medical waste, as source of stem cells for regenerative medicine. The immunological features of the fetal annexes, the phenotypical characterization, differentiation potential and secretomes of the stem cells derived from these annexes are described.

 No major weakness detected

 Specific comments

Please provide a more clear description of Chorionic stroma cells (hCSCs) vs. hAMSCs and add hCSCs to Table 1 and 2 

We agree with reviewer’s observation and modified the paper accordingly. In particular: 

-       We improved the manuscript explaining more in detail the differences between hCMSCs vs. hAMSCs. The sentence: 

“On the other side, hAMSCs can be obtained from the inner mesodermal tissue of amniotic membrane;” 

has been replaced with

“On the other side, hAMSCs can be obtained from the inner mesodermal tissue (the thicker basement membrane and collagen layer)of amniotic membrane

-       Moreover, we added at the end of the paragraph a new sentence:

Summarizing, two distinct types of stem cells can be isolated from amniotic membrane: the hAECs which derive from the epithelial part of amnion, and the hAMSCs, that can be isolated from the mesenchymal layer of amnios. On the other hand, hCMSCs can be isolated from the chorionic membrane. Thus, manually separating the amnios from the chorion after the delivery, it is possible to isolate the hAECs, hAMSCs or hCMSCs: these three populations differ for morphology, phenotype and differentiative potential (Table 1 and Table2)” 

-       hCMSCs has been added to Table 1 and 2. 

Add A and B to Fig.1 

We apologize for the inattention. The letters have been now added to the figure.